# Effectiveness of the 2024–2025 KP.2 COVID-19 vaccines in the United States during long-term follow-up

George N. Ioannou [1,2] ✉, Kristin Berry [1], Lei Yan[3,4], Yuan Huang[3,4], Hung-Mo Lin[3,4], David Bui[5], Denise M. Hynes [6,7], Edward J. Boyko [8], Jacqueline M. Ferguson [9], Mihaela Aslan[3,10] & Kristina L. Bajema [6,11]

Up-to-date estimates of COVID-19 vaccine effectiveness (VE) are needed to inform COVID-19 vaccination strategies and recommendations. This target trial emulation study aimed to estimate the long-term vaccine effectiveness (VE) of the 2024-2025 COVID-19 vaccines targeting the KP.2 Omicron variant within the Veterans Health Administration. The study population (90.9% male, mean age 70.7 years) included 538,631 pairs of vaccinated (i.e., received the KP.2 COVID-19 vaccine) and matched unvaccinated (i.e., did not receive the KP.2 COVID-19 vaccine) persons enrolled from August 2024 to January 2025. Over a mean follow-up of 172 days (range 97-232) extending to April 12, 2025, VE was low against laboratory-diagnosed SARS-CoV-2 infection (16.60%, 95% confidence interval [CI], 11.92-21.44), SARS-CoV-2-associated emergency department/urgent care (ED/UC) visit (21.05%, 95% CI, 14.22-27.21), SARS-CoV-2-associated hospitalization (19.53%, 95% CI 6.56-30.10) and much higher against SARS-CoV-2-associated death (65.53%, 95% CI 27.79-83.37). VE declined from 60 to 90 to 120 days against infection (31.28%, 25.81%, 22.44% respectively), ED/UC visit (34.40%, 29.19%, 25.71% respectively), hospitalization (37.39%, 28.98%, 22.52% respectively) and death (75.02%, 71.02%, 63.08% respectively). In conclusion, COVID-19 vaccines targeting the KP.2 variant used in the 2024-2025 season offered high protection against death and modest protection against infection, ED/UC visits or hospitalization, and VE declined over time.

In August 2024, the US Food and Drug Administration (FDA) approved updated 2024-2025 COVID-19 vaccines from Pfizer-BioNTech[1] and Moderna[2] targeting the KP.2 Omicron variant and authorized a vaccine from Novavax targeting the JN.1 Omicron variant[3]. The Centers for Disease Control and Prevention (CDC) recommended that all persons ≥6 months of age receive a 2024-2025 COVID-19 vaccine dose[4,5]. Despite this recommendation, only 13% of children and 23% of adults in the United States reported having received the 2024-2025 COVID-19 vaccine by the end of April 2025[6].

Real-world vaccine effectiveness (VE) studies of the earlier 2023-2024 XBB.1.5 COVID-19 vaccines estimated modest VE, which waned over a period of 6 months[7–9]. A test-negative, case-control study of the 2024–2025 Pfizer-BioNTech KP.2 vaccine in the Veterans Health Administration (VHA) estimated that early VE at a median 33 days after vaccination was 68% (95% CI 42–82%), 57% (95% CI 46–65%), and 56% (95% CI 36–69%) against COVID-19-associated hospitalizations, emergency department (ED) and urgent care (UC) visits, and outpatient visits, respectively[10]. Test-negative, case-control studies in two CDC networks estimated that the VE of the 2024-2025 COVID-19 vaccines at a median time of 55 days after vaccination was 33% (95% CI 28–38) against ED/UC visits among adults age ≥18 years and 45% (95% CI 36–53) against hospitalization among adults age ≥65 years[11]. However,

these studies estimated VE during a very short period after vaccination, and therefore do not provide data on longer term VE, which may wane over time. These studies were also not able to estimate VE against SARS-CoV-2-associated death, and, due to their case-control design, could not estimate cumulative incidence rates and differences in outcomes, which are needed to estimate the absolute number of persons needed to vaccinate to prevent an outcome.

VHA is one of the largest integrated health care systems in the United States. It provides care to more than 9 million Veterans, many of whom are older and have multiple comorbidities. VHA has utilized a national, comprehensive electronic health records (EHR) system for several decades, which provided adequate, high-quality EHR data for the execution of target trial emulation (TTE) studies of COVID-19 and RSV VE[7,12–16]. TTE involves specifying the critical study design elements of an RCT (eligibility criteria, treatment strategies, treatment assignment, time zero, outcomes, causal contrasts and analysis plan) and explicitly attempting to emulate these elements in an observational dataset[17,18]. We used TTE study design to specify and emulate a trial comparing KP.2 COVID-19 vaccination versus no KP.2 COVID-19 vaccination conducted among VHA enrollees aged ≥18 years with enrollment from 08/23/2024 until 01/17/2025 and follow-up through 04/12/2025. We aimed to estimate KP.2 COVID-19 VE against SARS-CoV-2 documented infection, ED/UC visit, hospitalization, and death during long-term follow-up, and describe the extent to which VE declined over time since vaccination.

## Results

### Study participant characteristics

The enrollment period of the emulated trial began on 08/23/2024, when KP.2 COVID-19 vaccines were FDA approved, and extended until 01/17/2025. During this overall enrollment period, 8 sequential trials were executed, each having an ~2-week enrollment period (Fig. 1)[19]. Key study design elements of the specified and emulated trials are compared in Supplemental Table 1. Of 2,119,853 VHA enrollees who fulfilled eligibility criteria, 541,213 received KP.2 COVID-19 vaccination during the enrollment period. Their characteristics before matching are shown in Supplemental Table 2. Of these individuals who received KP.2 COVID-19 vaccination, 538,631 (99.5%, including 212,124 Pfizer and 326,507 Moderna vaccine recipients) were matched to 538,631 unvaccinated (i.e., did not receive the KP.2 COVID-19 vaccine) person-trials (409,051 unique control individuals) (Fig. 2). Baseline characteristics were well-balanced between the two groups after matching, with all standardized mean differences (SMDs) below 0.10. Table 1 and Supplemental Figs. 1 and 2 show SMDs of baseline characteristics before and after matching. Matched individuals were 90.9% male, 66.5% White, 22.8% Black, 6.0% Hispanic/Latino, and had a mean age of 70.7 years. Most recent prior documented COVID-19 vaccination was within 90–182 days in 7.3%, 183–364 days in 57.0% and ≥365 days in 35.7%. The proportion who had ≥4 prior COVID-19 vaccines was 89.0%. The study population peaked in trial 2 (9/30/2024 to 10/13/2024, 122,777 individuals in each arm or 22.8%) and declined with each sequential trial, down to only 12,526 individuals (2.3%) in each arm in trial 8. Comorbidities such as diabetes (42.1%), coronary heart disease (37.7%), chronic kidney disease (30.0%) and congestive heart failure (12.7%) were common. Crossovers from the unvaccinated to the vaccinated arm after initial matched assignment occurred in 143,131 persons corresponding to 190,873 (35.4%) matched pairs.

### Vaccine effectiveness estimates and decrease over longer follow-up time

Over a mean follow-up of 172 days (range 97–232), there were 7863 laboratory-confirmed SARS-CoV-2 infections, 4353 SARS-CoV-2 associated ED/UC visits, 1046 SARS-CoV-2-associated hospitalizations, and 140 SARS-CoV-2-associated deaths, among 537,466 matched pairs of individuals who remained at risk 10 days following the index date (Figs. 3 and 4).

VE against laboratory-confirmed infection at 60 days was 31.28% (95% CI 25.56–36.36) and decreased progressively when estimated at

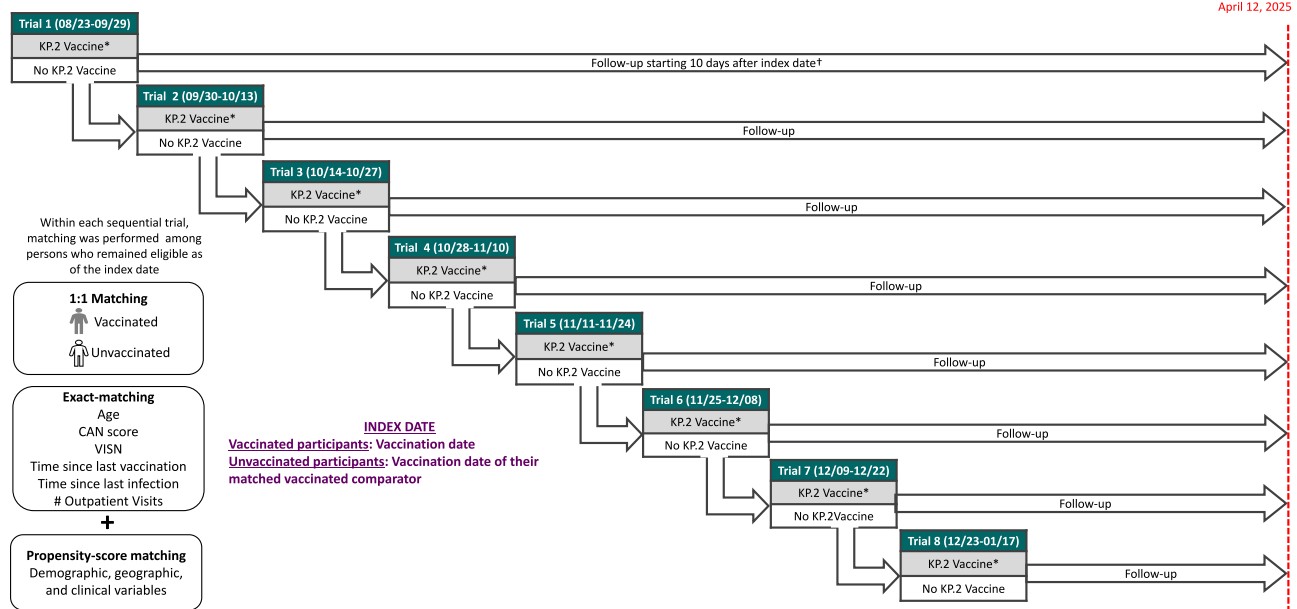

**Fig. 1 | Overview of the study design and matching approach using eight sequential trials, each with an approximately 2-week enrollment period, to emulate a target randomized controlled trial evaluating the effectiveness of the 2024-2025 KP.2 COVID-19 vaccine.** The combined enrollment period for the eight trials ran from 08/23/2024 to 01/17/2025, with follow-up extending through 04/12/2025. Outcome included vaccine effectiveness against documented SARS-CoV-2 infection, SARS-CoV-2 associated ED/UC visit, SARS-CoV-2 associated hospitalization or SARS-CoV-2 associated death. CAN Care Assessment Need, VISN Veterans Integrated Services Network. *Persons who received KP.2 vaccination during a given 2-week trial period were not eligible for later trials, whereas persons who remained unvaccinated could be included in subsequent periods if they continued to satisfy eligibility requirements. † Follow-up for outcomes started 10 days after the index date (the vaccination date or the same assigned date for the matched unvaccinated comparator) and continued through April 12, 2025.

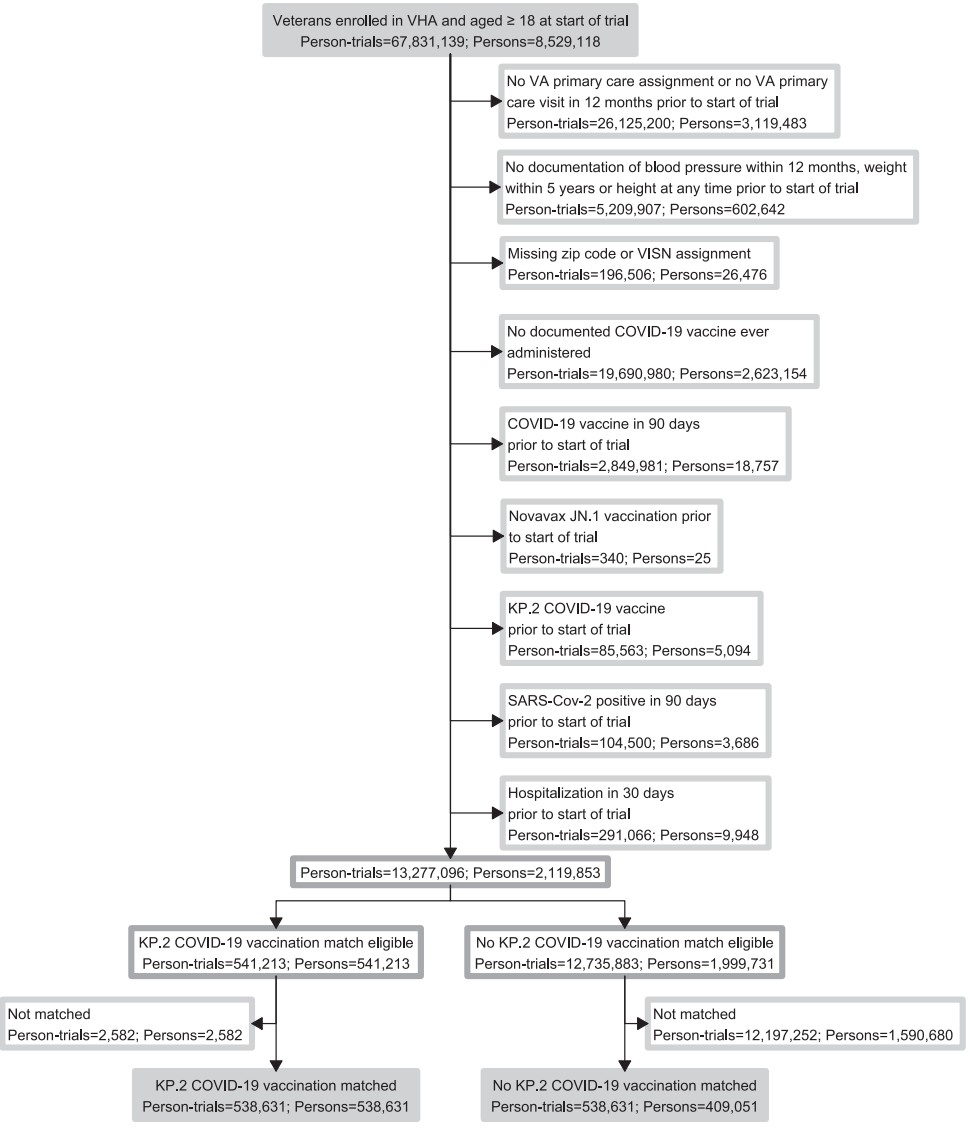

**Fig. 2 | Flow diagram showing derivation of the study population.** Identification of eligible individuals in the target trial emulation comparing KP.2 COVID-19 vaccination versus no KP.2 vaccination.

90 days (25.81%, 95% CI 20.90–29.75), 120 days (22.44%, 95% CI 17.65–26.01) or through the end of follow-up at day 232 (16.60%, 95% CI 11.92–21.44) (Fig. 4). Similarly, VE against SARS-CoV-2-associated ED/UC visit decreased progressively when extending to 60 days (34.40%, 95% CI 27.34–41.16), 90 days (29.19%, 95% CI 23.65–35.00), 120 days (25.71%, 95% CI 19.92–30.88) or the end of follow-up (21.05%, 95% CI 14.22–27.21). VE against SARS-CoV-2-associated hospitalization decreased when extending to 60 days (37.39%, 95% CI 20.83–49.61), 90 days (28.98%, 95% CI 15.19–40.51), 120 days (22.52%, 95% CI 10.02–31.72) or the end of follow-up (19.53%, 95% CI 6.56–30.10). VE against SARS-CoV-2-associated death decreased when extending to 60 days (75.02%, 95% CI 48.01–91.63), 90 days (71.02%, 95% CI 45.78–84.69), 120 days (63.08%, 95% CI 43.73–77.89) or the end of follow-up (65.53%, 95% CI 27.79–83.37).

The cumulative incidence of SARS-CoV-2 infection at the end of follow-up was 7.74/1000 (95% CI 7.37–7.97) in the vaccinated arm versus 9.29/1000 (95% CI 8.64–9.67) in the unvaccinated arm, resulting in risk difference of −1.54/1000 (95% CI −2.10 to −1.07), corresponding to 649 persons (95% CI 476–935) needed-to-vaccinate to prevent one infection (Supplemental Table 3). The cumulative incidence of SARS-CoV-2 associated ED/UC visit was 4.24/1000 (95% CI 3.98–4.43) in the vaccinated arm versus 5.36/1000 (95% CI 4.89–5.72) in the unvaccinated arm, resulting in risk difference of −1.13/1000 (95% CI −1.54 to −0.72), corresponding to 885 persons (95% CI 649–1389) needed-to-vaccinate to prevent one visit. The cumulative incidence of SARS-CoV-2 associated hospitalization was 0.97/1000 (95% CI 0.88–1.07) in the vaccinated arm versus 1.21/1000 (95% CI 1.04–1.38) in the unvaccinated arm, resulting in risk difference of −0.24/1000 (95% CI −0.40 to −0.07), which corresponded to 4167 persons (95% CI 2500–14,285) needed-to-vaccinate to prevent one hospitalization. The cumulative incidence of SARS-CoV-2 associated death was 0.11/1000 (95% CI 0.08–0.14) in the vaccinated arm versus 0.31/1000 (95% CI 0.14–0.64) in the unvaccinated arm, resulting in risk difference of −0.20/1000 (95% CI −0.47 to −0.04), which corresponded to 5000 persons (95% CI 2128–25,000) needed-to-vaccinate to prevent one death.

## Subgroup analyses, sensitivity analyses and negative outcome control

No clear trends in VE were observed across categories of CAN risk score or time since prior SARS-CoV-2 positive test (Supplemental Table 4). VE against infections, ED/UC visit and hospitalization was

**Table 1 | Baseline characteristics of veterans matched in an emulated target trial of KP.2 COVID-19 vaccination versus no KP.2 COVID-19 vaccination, with enrollment from 08/23/2024 until 01/17/2025 and follow-up extending through 04/12/2025[a]**

| | KP.2 COVID-19 vaccination (N = 538,631) | No KP.2 COVID-19 vaccination (N = 538,631) | Overall (N = 1,077,262) |
|---|---|---|---|
| **Mean age (SD), year** | 70.7 (11.6) | 70.6 (11.9) | 70.7 (11.7) |
| **Age group, no. (%)** | | | |
| 18–64 years | 131,907 (24.5%) | 131,907 (24.5%) | 263,814 (24.5%) |
| 65–74 years | 165,106 (30.7%) | 165,106 (30.7%) | 330,212 (30.7%) |
| ≥75 years | 241,618 (44.9%) | 241,618 (44.9%) | 483,236 (44.9%) |
| **Sex, no. (%)** | | | |
| Female | 50,112 (9.3%) | 48,287 (9.0%) | 98,399 (9.1%) |
| Male | 488,519 (90.7%) | 490,344 (91.0%) | 978,863 (90.9%) |
| **Race[b], no. (%)** | | | |
| American Indian, Alaska Native | 3683 (0.7%) | 3872 (0.7%) | 7555 (0.7%) |
| Asian | 7396 (1.4%) | 7763 (1.4%) | 15,159 (1.4%) |
| Black | 125,274 (23.3%) | 120,633 (22.4%) | 245,907 (22.8%) |
| Native Hawaiian, Other Pacific Islander | 4746 (0.9%) | 4898 (0.9%) | 9644 (0.9%) |
| White | 356,704 (66.2%) | 359,788 (66.8%) | 716,492 (66.5%) |
| Multiple | 4549 (0.8%) | 4622 (0.9%) | 9171 (0.9%) |
| Declined/Unknown/Missing | 36,279 (6.7%) | 37,055 (6.9%) | 73,334 (6.8%) |
| **Ethnicity[b], no. (%)** | | | |
| Hispanic/Latino | 33,728 (6.3%) | 31,355 (5.8%) | 65,083 (6.0%) |
| Not Hispanic/Latino | 480,672 (89.2%) | 482,707 (89.6%) | 963,379 (89.4%) |
| Declined/Unknown/Missing | 24,231 (4.5%) | 24,569 (4.6%) | 48,800 (4.5%) |
| **Rurality, no. (%)** | | | |
| Urban | 383,182 (71.1%) | 375,542 (69.7%) | 758,724 (70.4%) |
| Rural | 155,449 (28.9%) | 163,089 (30.3%) | 318,538 (29.6%) |
| **US geographical region[d], no. (%)** | | | |
| West | 124,408 (23.1%) | 124,408 (23.1%) | 248,816 (23.1%) |
| Midwest | 138,747 (25.8%) | 138,747 (25.8%) | 277,494 (25.8%) |
| Northeast | 99,353 (18.4%) | 99,353 (18.4%) | 198,706 (18.4%) |
| South | 176,123 (32.7%) | 176,123 (32.7%) | 352,246 (32.7%) |
| **Drive distance to nearest VHA medical facility in miles, no. (%)** | | | |
| 0–9 | 253,644 (47.1%) | 251,,015 (46.6%) | 504,659 (46.8%) |
| 10–24 | 198,489 (36.9%) | 196557 (36.5%) | 395,046 (36.7%) |
| 25–49 | 71,326 (13.2%) | 74,490 (13.8%) | 145,816 (13.5%) |
| ≥50 | 15,172 (2.8%) | 16,569 (3.1%) | 31,741 (2.9%) |
| **Drive time to nearest VHA medical facility in minutes, no. (%)** | | | |
| 0–9 | 122,010 (22.7%) | 120,184 (22.3%) | 242,194 (22.5%) |
| 10–19 | 225,450 (41.9%) | 221,743 (41.2%) | 447,193 (41.5%) |
| 20–59 | 179,942 (33.4%) | 184,480 (34.2%) | 364,422 (33.8%) |
| ≥60 | 11,229 (2.1%) | 12,224 (2.3%) | 23,453 (2.2%) |
| **Smoking, no. (%)** | | | |
| Current Smoker | 64,926 (12.1%) | 68,483 (12.7%) | 133,409 (12.4%) |
| Former Smoker | 255,978 (47.5%) | 257,571 (47.8%) | 513,549 (47.7%) |
| Never Smoker | 213,268 (39.6%) | 208,160 (38.6%) | 421,428 (39.1%) |
| Unknown | 4459 (0.8%) | 4417 (0.8%) | 8876 (0.8%) |
| **Alcohol Use Disorder[c], no. (%)** | 105,904 (19.7%) | 107,662 (20%) | 213,566 (19.8%) |
| **Substance Use Disorder[c], no. (%)** | 72,218 (13.4%) | 74,404 (13.8%) | 146,622 (13.6%) |
| **Body mass index, no. (%)** | | | |
| <18·5 kg/m² | 4900 (0.9%) | 5072 (0.9%) | 9972 (0.9%) |
| 18·5–24·9 kg/m² | 103,051 (19.1%) | 102,352 (19.0%) | 205,403 (19.1%) |
| 25–29·9 kg/m² | 196,027 (36.4%) | 195,373 (36.3%) | 391,400 (36.3%) |
| 30–34·9 kg/m² | 141,955 (26.4%) | 142,114 (26.4%) | 284,069 (26.4%) |
| 35–39·9 kg/m² | 62,033 (11.5%) | 62,904 (11.7%) | 124,937 (11.6%) |
| ≥40 kg/m² | 30,665 (5.7%) | 30,816 (5.7%) | 61,481 (5.7%) |
| **Chronic Kidney Disease[c], no. (%)** | 162,364 (30.1%) | 160,738 (29.8%) | 323,102 (30%) |
| **Diabetes[c], no. (%)** | 227,286 (42.2%) | 225,750 (41.9%) | 453,036 (42.1%) |
| **Coronary Heart Disease[c], no. (%)** | 202,855 (37.7%) | 203,439 (37.8%) | 406,294 (37.7%) |
| **Congestive Heart Failure[c], no. (%)** | 68,726 (12.8%) | 68,255 (12.7%) | 136,981 (12.7%) |

**Table 1 (continued) | Baseline characteristics of veterans matched in an emulated target trial of KP.2 COVID-19 vaccination versus no KP.2 COVID-19 vaccination, with enrollment from 08/23/2024 until 01/17/2025 and follow-up extending through 04/12/2025[a]**

| | KP.2 COVID-19 vaccination (N = 538,631) | No KP.2 COVID-19 vaccination (N = 538,631) | Overall (N = 1,077,262) |
|---|---|---|---|
| **Chronic Lung Disease[c], no. (%)** | 136,561 (25.4%) | 136,235 (25.3%) | 272,796 (25.3%) |
| **Dementia[c], no. (%)** | 17,250 (3.2%) | 17,803 (3.3%) | 35,053 (3.3%) |
| **Received immunosuppressive or cancer medications in the last year, no. (%)** | 28,538 (5.3%) | 27,561 (5.1%) | 56,099 (5.2%) |
| **Charlson Comorbidity Index, mean (SD)** | 2.5 (2.4) | 2.4 (2.4) | 2.4 (2.4) |
| **Care Assessment Need (CAN) score for 90-day mortality, no. (%)** | | | |
| 0–50 | 181,562 (33.7%) | 181,562 (33.7%) | 363,124 (33.7%) |
| 51–89 | 283,403 (52.6%) | 283,403 (52.6%) | 566,806 (52.6%) |
| 90–99 | 73,666 (13.7%) | 73,666 (13.7%) | 147,332 (13.7%) |
| **VHA primary care encounters in the last year, no. (%)** | | | |
| 0–1 | 37,989 (7.1%) | 36,161 (6.7%) | 74,150 (6.9%) |
| 2–3 | 120,901 (22.4%) | 119,062 (22.1%) | 239,963 (22.3%) |
| 4–7 | 176,825 (32.8%) | 178,751 (33.2%) | 355,576 (33.0%) |
| ≥8 | 202,916 (37.7%) | 204,657 (38.0%) | 407,573 (37.8%) |
| **VHA specialty care encounters in the last year, no. (%)** | | | |
| 0–1 | 18,256 (3.4%) | 17,955 (3.3%) | 36,211 (3.4%) |
| 2–3 | 59,438 (11.0%) | 58,517 (10.9%) | 117,955 (10.9%) |
| 4–7 | 129,944 (24.1%) | 129,845 (24.1%) | 259,789 (24.1%) |
| ≥8 | 330,993 (61.5%) | 332,314 (61.7%) | 663,307 (61.6%) |
| **VHA hospitalizations in the last year, no. (%)** | | | |
| 0 | 464,215 (86.2%) | 463,751 (86.1%) | 927,966 (86.1%) |
| 1 | 53,095 (9.9%) | 53,404 (9.9%) | 106,499 (9.9%) |
| 2–4 | 20,028 (3.7%) | 20,187 (3.7%) | 40,215 (3.7%) |
| ≥5 | 1293 (0.2%) | 1289 (0.2%) | 2582 (0.2%) |
| **Trial number and enrollment period, no. (%)** | | | |
| 1: 08/23/24 to 09/29/24 | 109,588 (20.3%) | 109,588 (20.3%) | 219,176 (20.3%) |
| 2: 09/30/24 to 10/13/24 | 122,777 (22.8%) | 122,777 (22.8%) | 245,554 (22.8%) |
| 3: 10/14/24 to 10/27/24 | 96,967 (18.0%) | 96,967 (18.0%) | 193,934 (18.0%) |
| 4: 10/28/24 to 11/10/24 | 77,086 (14.3%) | 77,086 (14.3%) | 154,172 (14.3%) |
| 5: 11/11/24 to 11/24/24 | 53,873 (10.0%) | 53,873 (10.0%) | 107,746 (10.0%) |
| 6: 11/25/24 to 12/08/24 | 34,048 (6.3%) | 34,048 (6.3%) | 68,096 (6.3%) |
| 7: 12/08/24 to 12/22/24 | 31,766 (5.9%) | 31,766 (5.9%) | 63,532 (5.9%) |
| 8: 12/23/24 to 01/17/25 | 12,526 (2.3%) | 12,526 (2.3%) | 25,052 (2.3%) |
| **Time since the most recent prior COVID-19 vaccine, no. (%)** | | | |
| 90–182 days | 39,519 (7.3%) | 39,519 (7.3%) | 79,038 (7.3%) |
| 183–364 days | 307,057 (57.0%) | 307,057 (57.0%) | 614,114 (57.0%) |
| ≥365 days | 192,055 (35.7%) | 192,055 (35.7%) | 384,110 (35.7%) |
| **Number of prior COVID-19 vaccination(s) no. (%)** | | | |
| 1–2 | 15,416 (2.9%) | 15,458 (2.9%) | 30,874 (2.9%) |
| 3 | 43,771 (8.1%) | 43,732 (8.1%) | 87,503 (8.1%) |
| ≥4 | 479,444 (89.0%) | 479,441 (89.0%) | 958,885 (89.0%) |
| **Time since the most recent prior laboratory-confirmed positive SARS-CoV-2 test, no. (%)** | | | |
| 90–182 days | 2987 (0.6%) | 2987 (0.6%) | 5974 (0.6%) |
| 183–364 days | 13,600 (2.5%) | 13,600 (2.5%) | 27,200 (2.5%) |
| ≥365 days | 114,346 (21.2%) | 114,346 (21.2%) | 228,692 (21.2%) |
| No prior laboratory-confirmed positive test | 407,698 (75.7%) | 407,698 (75.7%) | 815,396 (75.7%) |

[a]Baseline characteristics as of the start of each 2-week trial.
[b]Race and ethnicity were ascertained through a 2-question self-identified method included in the VHA Form 10-10EZ.
[c]Documented in the 2 years prior to the initiation of each 2-week trial.
[d]Regions are based on Veterans Integrated Services Networks (VISNs). West includes VISNs 19 to 22; Midwest includes VISNs 10, 12, 15, and 23; Northeast includes VISNs 1, 2, 4, and 5; and South includes VISNs 6–9, 16, and 17.

greater in persons who had >365 days since last vaccination than in those who had 90–365 days since last vaccination. There were no trends with increasing age in VE against infection, ED/UC visits or hospitalization; but VE against SARS-CoV-2-related death appeared to be limited to those ≥75 years of age (69.77%, 95% CI 29.00–85.72) with no effectiveness in those 18–74 years (11.77%, 95% −314.81 to 71.14), who had very low cumulative incidence of SARS-CoV-2 related death (0.05 per 1000 persons)

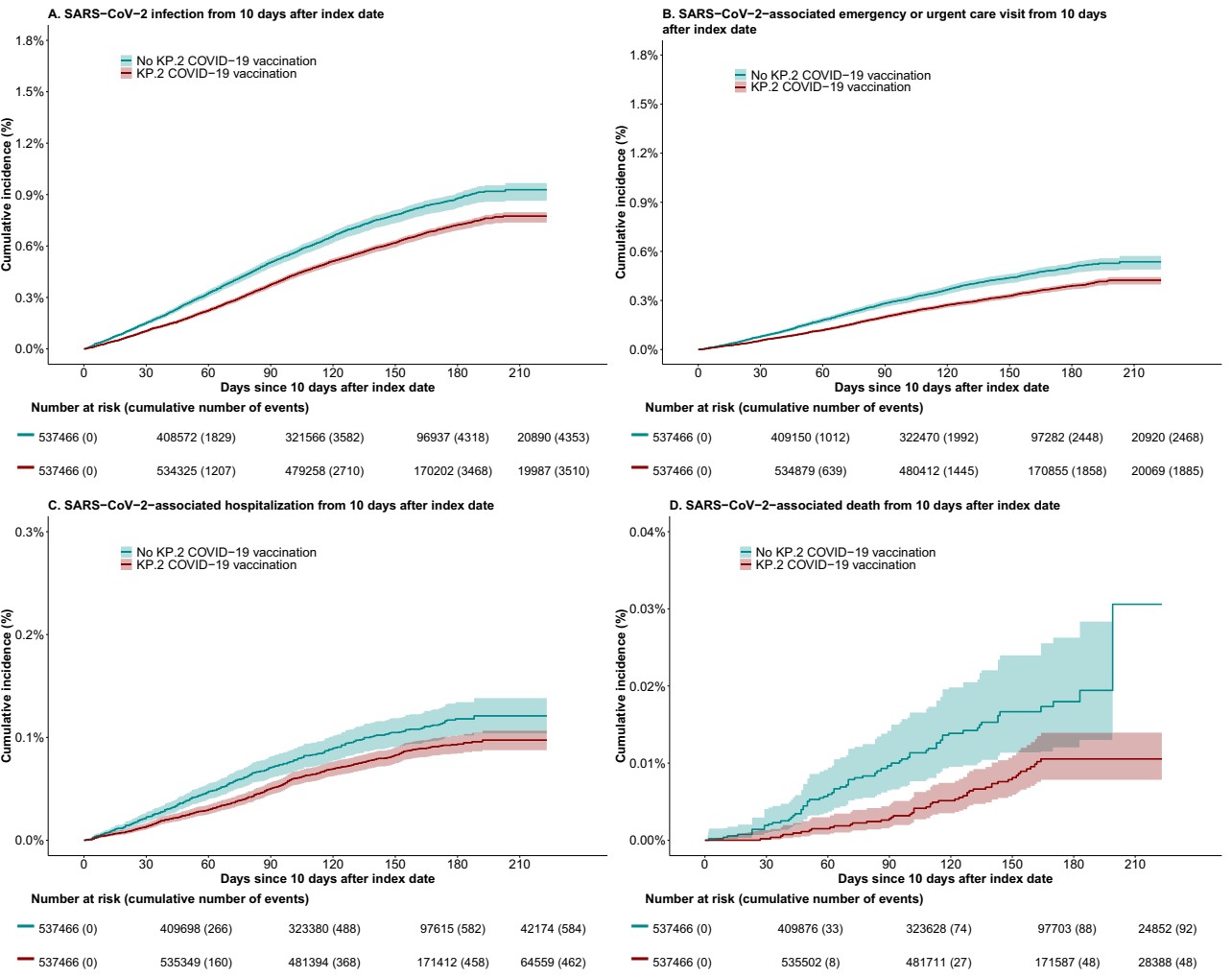

**Fig. 3 | Cumulative incidences of outcomes in vaccinated (i.e., received the KP.2 COVID-19 vaccine) versus matched unvaccinated (i.e., did not receive the KP.2 COVID-19 vaccine) individuals in a target trial emulation study extending from 08/23/2024 to 04/12/2025. A** documented SARS-CoV-2 infections. **B** SARS-CoV-2 associated ED/UC visits*. **C** SARS-CoV-2-associated hospitalizations*. **D** SARS-CoV-2-associated deaths†. Analysis is limited to 537,466 matched pairs both of whom remained uninfected and alive as of day 10 following the index date. The number of individuals at risk (and number of cumulative events) shown below the x-axis correspond to 0, 60, 120 180 and 232 days. The shaded area around the cumulative incidence lines illustrates the 95% confidence interval around the estimates of the cumulative incidence of documented SARS-CoV-2 infections (**A**), SARS-CoV-2 associated ED/UC visits (**B**), SARS-CoV-2-associated hospitalizations (**C**), and SARS-CoV-2-associated deaths (**D**). *ED/UC visit or hospitalization occurring from the day prior through 10 days after the eligible positive SARS-CoV-2 test result and having associated ICD-10 codes for acute respiratory infection. †Any death occurring the day of through 30 days after the eligible positive SARS-CoV-2 test result.

VE against ED/UC visit or hospitalization was slightly lower when we identified all ED/UC visits and hospitalizations within −1 to +10 days of a positive test (Supplemental Fig. 3), or ED/UC visits and hospitalizations that had associated ARI codes (Supplemental Fig. 4) rather than those that had associated COVID-19 codes. VE against SARS-Cov-2-associated ED/UC visit occurring within ±1 day of the test-positive date (Supplemental Fig. 5) was very similar to that determined within the prespecified window of −1 to 10 days.

VE in persons vaccinated in late trials (from 10/28/24 to 01/17/25) was not lower than VE in persons vaccinated in early trials (from 8/23/24 to 10/27/24) (Supplemental Table 5). Persons vaccinated in early trials tended to be slightly younger, with lower CAN score and longer interval since most recent prior COVID-19 vaccination (Supplemental Table 6).

Compared to the primary per-protocol analysis with IPCW, the per-protocol analysis with censoring of the matched pair at the time of vaccination of the unvaccinated comparator yielded very similar results (Supplemental Fig. 6) whereas the intention-to-treat analysis resulted in slightly lower VE estimates at all time points (i.e., 60-days, 90-say, 120-days and end of follow-up) against SARS-CoC-2 infection

(27.30%, 22.49%, 19.07%, 14.20% respectively), SARS-CoV-2-associated ED/UC visit (30.07%, 25.15%, 21.87%, 16.47%), SARS-CoV-2-associated hospitalization (33.72%, 26.18%, 20.04%, 15.44%) and SARS-CoV-2-associated death (71.48%, 65.98%, 55.77%, 46.20% respectively) (Supplemental Fig. 7).

There was little difference in the rate of all-cause ED/UC visits during days 1 through 9 after index date (the study's negative outcome control) in the vaccinated (cumulative incidence 14.73/1000) versus the unvaccinated arm (cumulative incidence 15.45/1000).

## Discussion

Our TTE study performed in the national VHA healthcare system between August 2024 and April 2025 found that during follow-up extending to 232 days from vaccination, the 2024–2025 COVID-19 vaccine targeting the KP.2 variant of Omicron had an estimated VE of 16.60% against laboratory-confirmed SARS-CoV-2 infection, 21.05% against SARS-CoV-2-associated ED/UC visit, 19.53% against SARS-CoV-2-associated hospitalization, and much higher VE of 65.53% against SARS-CoV-2-associated death. This corresponded to numbers-needed-

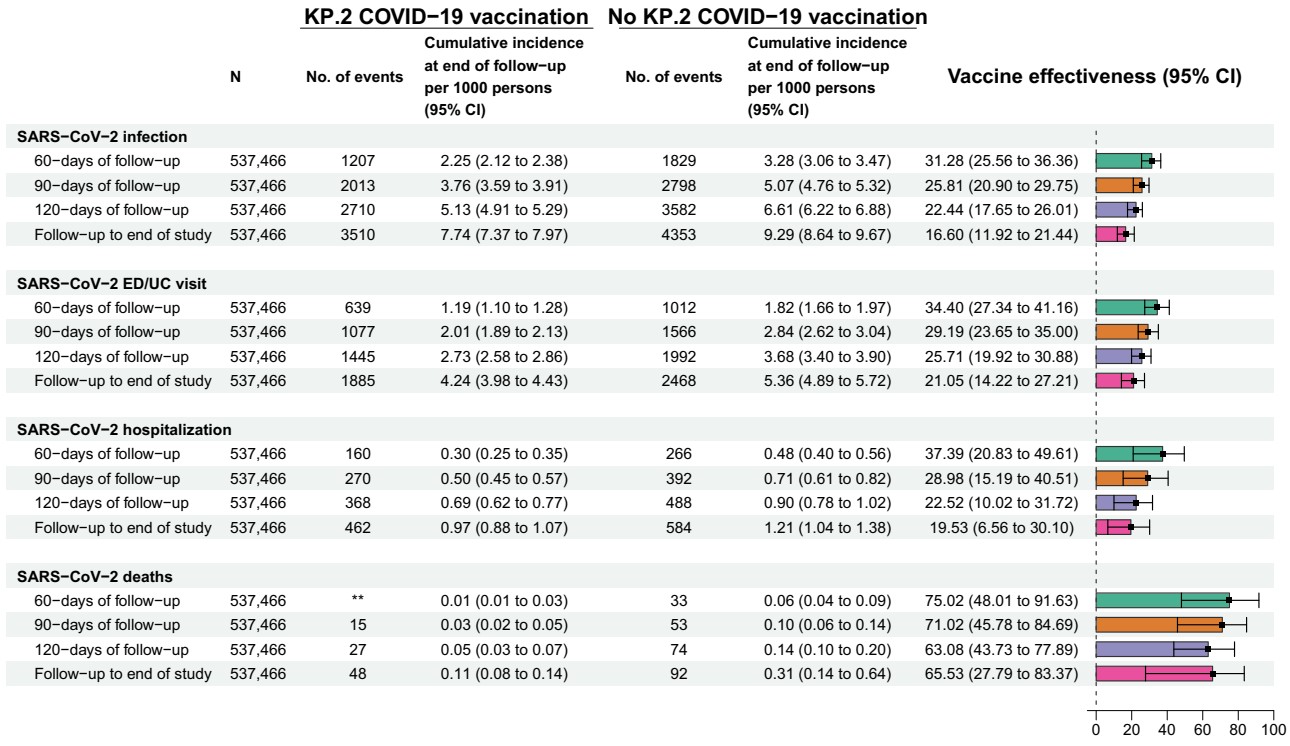

| | KP.2 COVID-19 vaccination | | | No KP.2 COVID-19 vaccination | | |
|---|---|---|---|---|---|---|
| | N | No. of events | Cumulative incidence at end of follow-up per 1000 persons (95% CI) | No. of events | Cumulative incidence at end of follow-up per 1000 persons (95% CI) | Vaccine effectiveness (95% CI) |
| **SARS-CoV-2 infection** | | | | | | |
| 60-days of follow-up | 537,466 | 1207 | 2.25 (2.12 to 2.38) | 1829 | 3.28 (3.06 to 3.47) | 31.28 (25.56 to 36.36) |
| 90-days of follow-up | 537,466 | 2013 | 3.76 (3.59 to 3.91) | 2798 | 5.07 (4.76 to 5.32) | 25.81 (20.90 to 29.75) |
| 120-days of follow-up | 537,466 | 2710 | 5.13 (4.91 to 5.29) | 3582 | 6.61 (6.22 to 6.88) | 22.44 (17.65 to 26.01) |
| Follow-up to end of study | 537,466 | 3510 | 7.74 (7.37 to 7.97) | 4353 | 9.29 (8.64 to 9.67) | 16.60 (11.92 to 21.44) |
| **SARS-CoV-2 ED/UC visit** | | | | | | |
| 60-days of follow-up | 537,466 | 639 | 1.19 (1.10 to 1.28) | 1012 | 1.82 (1.66 to 1.97) | 34.40 (27.34 to 41.16) |
| 90-days of follow-up | 537,466 | 1077 | 2.01 (1.89 to 2.13) | 1566 | 2.84 (2.62 to 3.04) | 29.19 (23.65 to 35.00) |
| 120-days of follow-up | 537,466 | 1445 | 2.73 (2.58 to 2.86) | 1992 | 3.68 (3.40 to 3.90) | 25.71 (19.92 to 30.88) |
| Follow-up to end of study | 537,466 | 1885 | 4.24 (3.98 to 4.43) | 2468 | 5.36 (4.89 to 5.72) | 21.05 (14.22 to 27.21) |
| **SARS-CoV-2 hospitalization** | | | | | | |
| 60-days of follow-up | 537,466 | 160 | 0.30 (0.25 to 0.35) | 266 | 0.48 (0.40 to 0.56) | 37.39 (20.83 to 49.61) |
| 90-days of follow-up | 537,466 | 270 | 0.50 (0.45 to 0.57) | 392 | 0.71 (0.61 to 0.82) | 28.98 (15.19 to 40.51) |
| 120-days of follow-up | 537,466 | 368 | 0.69 (0.62 to 0.77) | 488 | 0.90 (0.78 to 1.02) | 22.52 (10.02 to 31.72) |
| Follow-up to end of study | 537,466 | 462 | 0.97 (0.88 to 1.07) | 584 | 1.21 (1.04 to 1.38) | 19.53 (6.56 to 30.10) |
| **SARS-CoV-2 deaths** | | | | | | |
| 60-days of follow-up | 537,466 | ** | 0.01 (0.01 to 0.03) | 33 | 0.06 (0.04 to 0.09) | 75.02 (48.01 to 91.63) |
| 90-days of follow-up | 537,466 | 15 | 0.03 (0.02 to 0.05) | 53 | 0.10 (0.06 to 0.14) | 71.02 (45.78 to 84.69) |
| 120-days of follow-up | 537,466 | 27 | 0.05 (0.03 to 0.07) | 74 | 0.14 (0.10 to 0.20) | 63.08 (43.73 to 77.89) |
| Follow-up to end of study | 537,466 | 48 | 0.11 (0.08 to 0.14) | 92 | 0.31 (0.14 to 0.64) | 65.53 (27.79 to 83.37) |

■ 60-days of follow-up  ■ 90-days of follow-up  ■ 120-days of follow-up  ■ Follow-up to end of study

**Fig. 4 | Per protocol vaccine effectiveness estimates for the 2024–2025 KP.2 COVID-19 vaccine, evaluating protection against documented SARS-CoV-2 infection, SARS-CoV-2 associated ED/UC visits, SARS-CoV-2-associated hospitalization and SARS-CoV-2-associated death, during a study period extending from 08/23/2024 to 04/12/2025.** The error bars around vaccine effectiveness show the 95% confidence interval around the point estimate of vaccine effectiveness. **Omitted to meet reporting requirements when cell counts are less than eleven.

to-vaccinate of 649, 885, 4167, and 5000 to prevent each of the respective outcomes. VE declined from 60 to 90 to 120 days against infection (31.28%, 25.81%, 22.44% respectively), ED/UC visit (34.40%, 29.19%, 25.71% respectively), hospitalization (37.39%, 28.98%, 22.52% respectively), or death (75.02%, 71.02%, 63.08% respectively). These estimates provide useful data to inform recommendations for COVID-19 vaccinations[20].

Our KP.2 VE estimates are similar to those of another VHA-based study with a 6-month follow-up time, which compared simultaneous recipients of both COVID-19 and influenza vaccines to persons who received only the influenza vaccine[21]. Our estimates are substantially lower than those of another VHA-based study, which used a test-negative, case-control study design and reported VE against SARS-CoV-2-associated hospitalizations or ED/UC visits of 68% (95% CI 42–82%) and 57% (95% CI 46–65%) respectively[10]. However, median time since receipt of vaccination in that study was only 33 days. Test-negative, case-control VE studies from CDC networks estimated that VE against SARS-CoV-2-associated ED/UC visits at a median time of 55 days after KP.2 vaccination was 33% (95% CI 29–42%), which is very similar to our VE estimate at 60 days (34.40%, 95% CI 27.34–41.16), and 45% (95% CI 36–53) against hospitalization among adults age ≥65 years[11], which is slightly higher than our VE estimate at 60 days (37.39%, 95% CI 20.83–49.61). Our study extends these findings by demonstrating that VE continues to decrease over time such that over follow-up extending to 232 days, VE against SARS-CoV-2-associated ED/UC visit or hospitalization declined to 21% and 20% respectively.

One theoretical explanation for VE waning with time since vaccination might be the evolution of new predominant variants that are different than the KP.2 variant that the vaccine was designed to target. However, we did not find lower 60 and 90-day VE in persons vaccinated late (from October 2024 to January 2021) compared to persons vaccinated early (from August to October 2024) (Supplemental Table 5). This suggests that the observed waning VE over time since vaccination is unlikely to be related to the evolution of new predominant variants during the 2024-2025 respiratory virus season against which the KP.2 vaccine offered significantly less protection. This is also supported by the fact that the predominant variants after KP.2 (i.e., LB.1-like, KP.3.1.1-like, XEC, LP.8.1) descended from the same JN.1 lineage and had relatively low number of spike protein amino acid differences[22]. Other factors, such as waning levels of neutralizing antibodies, may account for the observed decrease in VE over time[23].

VE against SARS-CoV-2-associated death was substantially higher than VE against SARS-CoV-2-associated hospitalization or ED/UC visits, with follow-up extending to 60 days (75.02%, 95% CI 48.01–91.63), 90 days (71.02%, 95% CI 45.78–84.69), 120 days (63.08%, 95% CI 43.73–77.89) or the end of follow-up (65.53%, 95% CI 27.79–83.37). A recent VHA-based study estimated very similar VE against death of 64% (95% CI 23.0–85.8)[21]. This suggests that the KP.2 COVID-19 vaccine offers substantial protection against death if infected by SARS-CoV-2, which, combined with a modest protection against infection, resulted in a much more profound VE against death. These results are substantially higher than the VE against SARS-CoV-2-associated death (26.61%, 95% CI 5.53–42.32%) that we reported for the 2023–2024 COVID-19 vaccine that targeted the XBB.1.5 variant using VHA data and similar study design and follow-up period[7]. This reinforces the need to continue to assess VE for each updated vaccine. Our subgroup analyses suggested that the protection against SARS-CoV-2-associated death provided by the KP.2 vaccine during the 2024–2025 season was limited to older persons ages ≥75 years. In persons under 75 years of age, the cumulative incidence of SARS-CoV-2 related death was very low (0.05 per 1000 persons) and not substantially different between the vaccinated and unvaccinated groups.

In contrast to the test-negative, case-control study design, TTE enables estimation of cumulative incidences in vaccinated and matched unvaccinated arms, which can be used to estimate risk differences and number-needed-to-vaccinate to prevent an adverse outcome. We estimated the number-needed-to-vaccinate to prevent one documented SARS-CoV-2 infection (n = 649, 95% CI 476–935), ED/UC visit (n = 885, 95% CI 649–1389), hospitalization (n = 4167, 95% CI 2500–14,285), or death (n = 5000, 95% CI 2128–25,000). These numbers are larger than estimates of number-needed-to-vaccinate prevent one hospitalization against COVID-19 in the 2021-2022 season (n = 205, range 44–615)[24], or against influenza in the 2017–2018 season (n = 1223, 95% CI 578–3488)[25]. However, numbers-needed-to-vaccinate fluctuate based on disease incidence, vary in different risk groups (which our sample size precluded us from estimating), and do not capture the indirect effects of vaccines on unvaccinated persons (i.e., herd immunity)[26]. Hence, they should be interpreted with caution.

The COVID-19 vaccination strategy over the last 3 years has been to introduce an updated COVID-19 vaccine at the beginning of each respiratory virus season that targets the predominant Omicron variants circulating just prior to their introduction: BA.4/BA.5 (as part of the bivalent vaccines that also targeted the ancestral strain) for 2022–2023, XBB.1.5 for 2023–2024, and KP.2 for 2024–2025. Following the same rationale, in May 2025, FDA's Vaccines and Related Biological Products Advisory Committee (VRBPAC) recommended that the 2025–26 vaccines should target the JN.1 lineage and specifically the LP.8.1 sublineage[27]. FDA approved the JN.1/LP.8.1-targeting vaccines in August 2025 for adults ages 65 years and older as well as for individuals ages 5 through 64 with at least one high-risk condition. Although VE of the 2025–26 vaccine formulations cannot be predicted, our results of relatively low VE, coupled with very low vaccine uptake rates, call for accelerated efforts to develop new vaccination strategies that could provide higher and more sustained protection, such as mucosal vaccines effective against all variants including those presumed to develop in the future and as the virus continues to mutate.

Limitations of the study include, first, the fact that outcomes were anchored around laboratory-confirmed SARS-CoV-2 infections. Results of at-home COVID-19 antigen tests or infections that were not confirmed by any testing were not included. However, we anticipate that the vast majority SARS-CoV-2 infections that resulted in ED/UC visits, hospitalizations or deaths would have been confirmed by laboratory testing. Second, ascertainment of KP.2 COVID-19 vaccination and SARS-CoV-2-related outcomes might be incomplete. To ameliorate this, we accessed vaccinations administered both within and outside VHA and limited the study population to VHA enrollees who had previously received COVID-19 vaccination through VHA, had an assigned VHA primary care provider, and had documentation of recent VHA care. Third, although exact and propensity score matching achieved good balance across key potential confounders, some residual confounding may persist, including variation in healthcare seeking behavior. Nevertheless, our negative outcome control—ED/UC visits during days 0–9—showed little evidence of residual confounding or differential outcome ascertainment. Fourth, the reduction in estimated VE at progressively longer follow-up times may be at least in part due to a depletion-of-susceptibles bias[28] rather than true waning in effectiveness. Fifth, VHA enrollees included in our TTEs were predominantly male with advanced age and frequent comorbid conditions; our results should be extrapolated to other populations with caution.

In conclusion, VE of the KP.2-targeted COVID-19 vaccines during the 2024–2025 season was modest against SARS-CoV-2 infection, ED/UC visit and hospitalization and declined over time but was much higher against SARS-CoV-2 associated death with less evidence of waning over time, in an older population with multiple comorbidities. Continued real-world COVID-19 VE studies to assess the degree and durability of vaccine protection will be important to inform vaccine policy and motivate more effective and enduring vaccine strategies.

## Methods

### Ethics approval
Our study complies with all relevant ethical regulations and was reviewed by the VA Puget Sound Health Care System (VAPSHCS) Institutional Review Board (IRB) and approved by the VAPSHCS Research and Development Committee. As a retrospective study of existing electronic health records, the study was granted an expedited review and was verified to meet one or more of the exemption categories. As such, informed consent was not required and a waiver of HIPAA authorization was obtained.

### Specification and emulation of sequential target trials
Supplemental Table 1 outlines the key study design elements of the specified and emulated trials. The emulated trial's enrollment period began on 08/23/2024, the date of FDA approval of the KP.2 COVID-19 vaccines, and extended until 01/17/2025, following which vaccine administrations dwindled to very low numbers. During this interval, we implemented 8 sequential trials, each with an ~2-week enrollment period, to account for multiple potential enrollment times for individuals who had not yet received KP.2 COVID-19 vaccination and to allow ascertainment of updated eligibility and baseline characteristics for each interval (Fig. 1)[19]. Among persons who fulfilled eligibility criteria in each 2-week trial, we matched with replacement each person who received KP.2 vaccination during the 2-week period to one person who remained unvaccinated until the end of the 2-week period. We used exact matching followed by propensity score matching, to approximate the covariate balance that would be expected under stratified randomization. For each resulting matched pair, the unvaccinated individual was assigned the vaccination date of the matched vaccinated individual as their index date; both members of the pair were required to meet eligibility on that date. Follow-up began on the index date and continued until 04/12/2025. Individuals who received a KP.2 COVID-19 vaccine within a given 2-week trial window were excluded from consideration in later trials, whereas eligible persons who remained unvaccinated could be included in subsequent trials. (Fig. 1).

### Data sources
We drew upon the VHA Corporate Data Warehouse (CDW) and the COVID-19 shared data resource (CSDR), which compile near-real-time, EHR data from all VHA sites. These sources contain detailed demographic, geographic and clinical information, as well as records of ED/UC visits and hospital admissions. They also capture laboratory-confirmed SARS-CoV-2 test results and COVID-19 vaccinations administered within VHA or documented in the VHA EHR from external sources. The VHA contributes to the CDC's Immunization Gateway, which supplies vaccination records from state and local immunization information systems, as well as public and private vaccination partners. These vaccination records were available in CDW[29].

Centers for Medicare & Medicaid Services (CMS) data through May 2024 were incorporated to enhance ascertainment of vaccination history and prior SARS-CoV-2 infection status, although CMS data capturing outcomes during the study follow-up period were not available. Information from VHA-purchased community care encounters was additionally used to supplement baseline comorbidities, vaccination status, ED/UC visits, and hospitalizations.

### Eligibility criteria and study population
Eligible individuals included VHA enrollees ≥18 years of age who were assigned to a VHA primary care team and had a primary care visit in the 12 months prior to the beginning of each 2-week trial. Additionally, eligible individuals had documentation of blood pressure within 12 months, weight within 5 years and height at any time prior to each trial. Eligibility was further limited to those with documented residential address, an assigned VHA Integrated Service Network (VISN, VHA's 18 administrative regions[30]), and evidence of having received at

least one, earlier-formulation COVID-19 vaccine within the VHA system at any time in the past. These criteria were intended to ensure inclusion of persons actively engaged in VHA care and with prior access to VHA vaccination (Fig. 2).

We excluded individuals who had received any COVID-19 vaccine within the preceding 3 months or a Novavax JN.1 vaccine or a KP.2 COVID-19 vaccine at any time prior to the start of each 2-week trial, those who tested positive for SARS-CoV-2 in the previous 3 months, and those who had an inpatient hospitalization within 30 days before the beginning of each trial.

### Matching of unvaccinated to vaccinated individuals

In each 2-week trial, we began by exact-matching every eligible individual who received a KP.2 COVID-19 vaccine during that period to all eligible comparators who remained unvaccinated on the corresponding index date. To reduce the likelihood of failure to match, we limited the exact-matching variables to a small set of characaeristics[31], known to be strongly associated with both vaccination uptake and COVID-19 related outcomes among VHA enrollees[32–35]. These included: age category (18–64, 65–74, ≥75 years); Care Assessment Needs (CAN) score[36] category (0–50, 51–89, ≥90), a validated VHA-derived predictor of 90-day mortality automatically generated within the EHR[37]; VISN; time since most recent COVID-19 vaccine (90–182, 183–364 and ≥365 days prior to the beginning of each 2-week trial); time since most recent SARS-CoV-2 positive test (90–182, 183–364 and ≥365 days or no documented infection prior to the beginning of each 2-week trial), and number of primary and specialty outpatient healthcare visits (0–4, 5–8, 9–15, ≥16) (Supplemental Table 1).

Within each exact-matched stratum, we then applied a second matching step using propensity scores, allowing matches to be made with replacement. This step was intended to further mitigate residual confounding and to identify the closest possible comparator for each vaccinated individual. The propensity score model—constructed using logistic regression to estimate the probability of receiving vaccination—incorporated 23 demographic, geographic, health care utilization, and clinical variables that were prespecified based on their known association with both vaccination likelihood and study outcomes (Supplemental Table 1). For each vaccinated individual, we selected for matching the unvaccinated person with the closest propensity score, provided the score was within 0.2 standard deviations of the mean (SDM). To serve as a valid comparator, the unvaccinated individual had to continue meeting eligibility criteria on the matched index date—specifically, they needed to be alive, still unvaccinated, and without evidence of a prior positive SARS-CoV-2 test. This 2-step matching process was repeated 8 times for each sequential trial, with a distinct propensity score model generated for each 2-week period.

All covariates were determined based on their updated values as of the beginning of each 2-week trial. Declined, unknown or missing race and ethnicity (6.8% and 4.5% respectively) were treated as separate categories because they represent meaningful groups[37–39].

### Outcomes

The study evaluated VE against four primary outcomes. SARS-CoV-2 infection was defined as the first positive, laboratory-confirmed SARS-CoV-2 test (nucleic acid amplification or antigen test) obtained from a respiratory specimen beginning 10 days after the index date—an interval selected because little meaningful protection is expected earlier—and extending through the end of the follow-up period. SARS-CoV-2 associated urgent care (UC) or emergency department (ED) visit was defined as an ED/UC visit within 1 day before or 10 days after the eligible positive SARS-CoV-2 test together with documentation of ICD-10 codes for COVID-19 (U07.1, U07.2, J12.82, M35.81, Supplemental Table 6). SARS-CoV-2 associated hospitalization was defined as hospitalization within 1 day before or 10 days after the eligible positive SARS-CoV-2 test together with documentation of inpatient ICD-10

codes for COVID-19. In sensitivity analyses we identified all ED/UC visits or hospitalizations within −1 to +10 from the positive test, or those with associated codes for acute respiratory infection (ARI, Supplemental Table 7)[10] (rather than limiting to those with codes for COVID-19); or we limited to ED/UC visits and hospitalizations occurring within ±1 day of the positive SARS-CoV-2 test. SARS-CoV-2 associated death was defined as death from any cause within 30 days after the qualifying positive SARS-CoV-2 test.

### Statistical analysis

We evaluated covariate balance after matching using standardized mean differences (SMDs)[40]. The primary analytic cohort was limited to matched pairs in which neither person had a positive SARS-CoV-2 test during days 0–9 following the index date and both were alive at day 10, when follow-up commenced. Analyses were conducted with matched pairs followed up for 60 days (i.e., from day 10 to 70 after index date), 90, 120 or until the end of follow-up on 4/12/2025 (maximum follow-up 232 days). We used a cause-specific Aalen-Johansen estimator which accounts for the presence of death as a competing risk in calculating the cumulative incidences. VE was calculated as $100 \times (1 − \text{risk ratio})$, with the risk ratio defined as the ratio of Aalen-Johansen cumulative incidence estimates over the specified follow-up period. Non-SARS-CoV-2-associated deaths were treated as a competing risk. VE confidence intervals (95%) were estimated by using a nonparametric percentile blocked bootstrap with 500 repetitions.[41] Risk difference was defined as the difference in cumulative incidence estimates, and its 95% confidence intervals were similarly derived via bootstrap methods.

Anticipating a substantial rate of cross-over from the unvaccinated to the vaccinated arm, we performed a per-protocol analysis as the primary analyses, where unvaccinated comparators were censored at the time of crossover with application of inverse probability of censoring weights (IPCW) on a bi-weekly basis[42,43]. We used the same baseline and time-varying covariates included in the matching process as well as calendar time since the start of the study on August 23, 2024 and calendar time squared, updated bi-weekly, with weights estimated using pooled logistic regression for matched unvaccinated comparators, while matched vaccinated persons had a censoring weight of 1. We performed two additional sensitivity analyses: a per-protocol analysis where matched pairs were censored at the time of crossover of the unvaccinated individuals and an intention-to-treat analysis, in which matched unvaccinated individuals who were later vaccinated after the index date remained analyzed in the unvaccinated group.

We compared as a negative outcome control[44] the incidence of all-cause ED/UC visits during days 1–9 days after index date, which should not be affected by vaccination, to assess for residual confounding or outcome ascertainment bias. In prespecified subgroup analyses, VE was estimated by age, CAN score, time since last vaccination and time since last infection, based on the categories used for exact matching. We compared 60-day and 90-day VE in the late trials (with enrollment from 10/28/24 to 01/17/25) versus the early trials (with enrollment from 8/23/24 to 10/27/24) to investigate whether any VE waning with time since vaccination might be explained by VE waning by calendar time.

All analyses were conducted using R software (version 4.4.1) and packages MatchIt (version 4.5.5), survival (version 3.7-0), boot (version 1.3-31), and speedglm (version 0.3-5)[45].

### Reporting summary

Further information on research design is available in the Nature Portfolio Reporting Summary linked to this article.

## Data availability

The data supporting the findings of this study are not publicly available due to the inclusion of identifiable protected health information from the Veterans Health Administration. Privacy regulations prevent the open sharing of the individual-level data used in this study and any

data covered under these regulations cannot be shared. The Veterans Health Administration may approve the sharing of some study data after verifying de-identification, though this may not include all final study data. Each request is subject to approval by the ethics board, privacy office, and information systems and security office. For such requests, please contact the corresponding author.

## Code availability

The code is available from the corresponding author upon request.

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

## Acknowledgements

This study was supported by the US Department of Veterans Affairs Cooperative Studies Program (CSP). The US Department of Veterans Affairs CSP provided general policies for the appropriate conduct of research according to ethical and scientific standards. Otherwise, it did not specify the design and conduct of the study; collection, management, analysis, and interpretation of the data; preparation, review, or approval of the manuscript; or decision to submit the manuscript for publication. The contents of this article do not represent the views of the US Department of Veterans Affairs or the US government. The study was supported by the US Department of Veterans Affairs Health Services Research & Development (HSR&D) grant C19 21-278 to GNI; HSR&D grant C19 21-279 to DMH; HSR&D RCS 21-136 grant to DMH. We thank the Biomedical Advanced Research and Development Authority (BARDA, IAA#: AAI21050) and US Food and Drug Administration (FDA, IAA#: 75F40121S30013) for their support.

## Author contributions

Conception and design of the study: G.N.I., K.L.B. (Kristina Bajema), K.B. (Kristin Berry), M.A., Y.H., D.B., L.Y., H.-M.L. Drafting of manuscript: G.N.I. Data generation: K.B. (Kristin Berry). Analysis and/or interpretation of the data: K.B. (Kristin Berry), G.N.I., K.L.B. (Kristina Bajema), D.B., L.Y. Statistical expertise: K.B. (Kristin Berry), L.Y., H.-M.L., Y.H., M.A. Critical revision of the manuscript for important intellectual content: all authors. Final approval of the article: all authors. Obtaining funding: G.N.I., K.L.B., D.M.H., M.A.

## Competing interests

The authors declare no competing interests.

## Additional information

[1]Research and Development, Veterans Affairs Puget Sound Health Care System, Seattle, WA, USA. [2]Divisions of Gastroenterology, Veterans Affairs Puget Sound Health Care System and University of Washington, Seattle, WA, USA. [3]Veterans Affairs Cooperative Studies Program Clinical Epidemiology Research Center (CSP-CERC), Veterans Affairs Connecticut Healthcare System, West Haven, CT, USA. [4]Department of Biostatistics, Yale School of Public Health, New Haven, CT, USA. [5]Veterans Affairs Portland Health Care System, Portland, OR, USA. [6]Center of Innovation to Improve Veteran Involvement in Care (CIVIC), Veterans Affairs Portland Healthcare System, Portland, OR, USA. [7]Health Management and Policy, College of Health and Health Data and Informatics Program, Center for Quantitative Life Sciences, Oregon State University, Corvallis, OR, USA. [8]Seattle Epidemiologic Research and Information Center, Veterans Affairs Puget Sound Health Care System, Seattle, WA, USA. [9]Center for Innovation to Implementation (Ci2i), VA Palo Alto Health Care System, Palo Alto, CA, USA. [10]Department of Medicine, Yale School of Medicine, New Haven, CT, USA. [11]Division of Infectious Diseases, Department of Medicine, Oregon Health and Science University, Portland, OR, USA. ✉e-mail: george.ioannou@va.gov

