## [Transparent Peer Review file · Nature Communications]

Effectiveness of the 2024-2025 KP.2 COVID-19 Vaccines in the United States During Long-Term Follow-Up

Corresponding Author: Dr George Ioannou

Version 0:

Reviewer comments:

Reviewer #1

(Remarks to the Author)

This target trial emulation study estimates the VE of the KP.2 vaccination against infection, ED/UC visit, hospitalization and death. The estimated effectiveness against non-mortality outcomes is generally low, and lower than other estimates, including one from the same health system, though it studied a shorter time period. The manuscript is generally well written and the methodology generally appropriate to the chosen study design.

I do have one major and several minor some comments:

Major comment:

The outcomes being events within a specific time frame of a positive test, without any indication that the encounter was due to COVID-19 or even somewhat associated with respiratory illness is a concern. Our own work has shown that VE against hospitalizations verified to be due to COVID is generally somewhat higher than VE against hospitalizations with a code for acute respiratory infection, but not necessarily due to COVID. This study including encounters regardless of cause is likely to dilute the effect of vaccination by adding noise to the data. I suspect that this is as much of a reason for the lower VE in this study compared to TND studies using ARI-associated encounters as differences in follow-up time.

Minor comments:

Abstract: What time point was used for the Aalen-Johansen estimates to calculate the overall VE? It is presented as “Over a mean follow-up of 172 days (range 97-232), VE was low...” which makes it seem like it would be at 172 days, but it seems more likely to be at 232 days, the end of follow-up. Please clarify.

This point comes up again in the results, line 115: “through the end of follow-up (16.60%, 95% CI 11.92-21.44), corresponding to a mean follow-up of 172 days.” Is the VE estimate from the Aalen-Johansen estimates at end of follow-up (232 days) or mean follow-up (172 days)? If this is just a place to list the mean follow-up time, I'd move that up to the participant characteristics paragraph.

Lines 125-126: It should be the number needed to vaccinate to prevent one infection, not one visit.

Lines 142-143: I would say that there's no statistically significant effectiveness against death in the younger age group, though the number of events in this group is really too small to be able to say anything definitive – the CI includes the VE estimate for those age 75+. Also, the lower CI of -314.81 seems a bit extreme – can the authors please verify it? If it is correct, what use is a VE estimate with a CI nearly 400% wide?

Lines 170-173: With the VHA-based TND study following patients through November (about 90 days since initial vaccination), I would expect their VE to be very similar to the 60-day estimates from this study – median time since vaccination in a TND study is not the same as follow-up time in a TTE. Differences this large seem more likely to be due to some bias due to the study methodology (like the aforementioned lack of requirement that the encounter be associated with COVID or ARI) or perhaps differences in VE against different strains circulating at different times of year. A sensitivity analysis stratifying by enrollment period could potentially show if VE is waning by calendar time as well as time since vaccination.

Lines 205-208 seem to suggest that, because this vaccine was not terribly effective, next year's vaccine also won't be. That's a bit of a stretch, though I do agree that the sum total of current COVID—19 vaccination research does suggest that next year's vaccine will also wane in effectiveness over time, so an updated vaccine technology may be appropriate to address this general shortcoming of current vaccines.

In the limitations I would suggest adding potential generalizability concerns, as VHA members are significantly different from the general population.

In Figure 3 the time when the number at risk and cumulative incidence are presented below the plots is not labeled. I assume that it is 0, 60 120, 180, and maybe 240 (or 232) days. I'd appreciate a label or footnote to confirm.

I am unsure of the journal's policy on significant digits for estimates, but two places after the decimal for VE estimates seems overly precise – I think one place after the decimal should be sufficient.

Reviewer #2

(Remarks to the Author)

The authors report a study, designed using the TTE framework, on COVID-19 vaccine effectiveness. The study is well performed and described. We have a few points of feedback.

We commend the authors on reporting the Number Needed to Vaccinate, this is a very intuitive measure for interpretation. Throughout the paper, the authors state that they study 'long-term follow-up'. While this is a subjective measure, we would not consider less than a year 'long-term'.

It is a surprising result that the estimated VE against SARS-CoV-2 associated hospitalization is much lower than against SARS-CoV-2 associated death. Most studies find a lower VE against infection, but VE against hospitalization and death are usually both high. As all cause hospitalizations were included, could the authors report something about the registered causes of the included SARS-CoV-2 associated hospitalizations, or, preferably, run some sensitivity analyses?

We would have chosen the per-protocol analysis with censoring of the pair when the unvaccinated gets vaccinated as primary analysis, as this prevents part of the bias due to depletion of susceptibles, at least it keeps the vaccinated and unvaccinated in covariate balance. However, as the results hardly differ, we assume the adjustment in the primary analysis suffices.

How were positive tests within the first 10 days after the index date handled? Was the pair censored?

Line 247: How likely is it that a VHA enrollee received a vaccination outside a VHA facility and how often are these recorded in the VHA EHR? As we read that older data were supplemented with CMS data. Can you say anything about the completeness based on past data?

Minor points:

Line 64: from what age are children recommended vaccination?

Line 74-75: "However, these studies estimated VE during a very short period after vaccination, which may overestimate VE over longer follow-up, if VE wanes over time." we would not call this an overestimation, it just estimates the short term VE which differs from the long term VE.

Line 81: a brief explanation of the TTE framework, with a reference, would be suitable here.

Line 85: "COVID-19 documented infection" I suggest to consistently use SARS-CoV-2 infection, as in the abstract 170 onward: is there any international literature on KP.2 VE?

180: we would not state as strongly that there was less waning, as the uncertainty is high.

188-189: was KP2 the dominant variant throughout the study period? How might this have influenced VE?

219-220: besides missing at-home antigen test results, many infections will have remained untested, which can be added here as a limitation.

311: can you shortly elaborate on how the competing risk was incorporated in the analysis?

Supplemental Table 1: How does reconfirming the matching criteria work? What happens if the match created at baseline is not an exact match at the index date anymore, or if the propensity scores are not the most similar anymore?

"waning vaccinations after December"; the use of waning may be confusing here.

Propensity scores: do you expect the number of prior COVID vaccinations to be a confounder?

Primary outcomes, right panel: For outcome 1, should this have the same text with 'randomization date' replaced by 'index date'?

Very minor:

103: participant > participants

260 withing > within

275 measure of 90-day mortality risk

315 form > from

318: as well as

Fig1: April 12, 2024 > April 12, 2025

Reviewer #3

(Remarks to the Author)

I co-reviewed this manuscript with one of the reviewers who provided the listed reports. This is part of the Nature Communications initiative to facilitate training in peer review and to provide appropriate recognition for Early Career

Researchers who co-review manuscripts.

Version 1:

Reviewer comments:

Reviewer #1

(Remarks to the Author)

I would like to thank the authors for their work in updating the manuscript. They have appropriately addressed my concerns.

I am surprised that many of the point estimates for VE in the early-season vaccine recipients is lower than that for the late-season recipients. I wonder if this could be due to differences in the vaccinated population, with higher-risk and immunocompromised individuals more likely to be vaccinated shortly after the vaccine is available.

Reviewer #2

(Remarks to the Author)

We are happy with the revision and recommend it for publication.

Reviewer #3

(Remarks to the Author)

Effectiveness of the 2024-2025 KP.2 COVID-19 Vaccines During Long-Term Follow-Up.

We would like to thank the Editor and Reviewers for their thoughtful comments. The major changes in the analyses performed in response to the comments were:

- We used as primary outcomes ED/UC visits and hospitalizations that occurred within -1 to +10 days from a positive SARS-CoV-2 test and had associated codes for COVID-19. In additional sensitivity analysis we evaluated as outcomes:
 - ED/UC visits and hospitalizations within -1 to +10 days from the positive SARS-CoV-2 test that had associated codes for acute respiratory infection
 - All ED/UC visits and hospitalizations within -1 to +10 days from the positive SARS-CoV-2 test.
 - ED/UC visits and hospitalizations within ± 1 day of the positive SARS-CoV-2 test that had associated COVID-19 codes
- We performed subgroup analysis stratifying by enrollment period from 8/23/24 to 10/27/24 (the first three sequential trials) versus 10/28/24 to 01/17/25 (the last five sequential trials) to evaluate whether the observed waning in VE over time could be explained by differences in VE against strains circulating at earlier versus later calendar times.

Below is a point-by-point response.

Reviewer #1:

This target trial emulation study estimates the VE of the KP.2 vaccination against infection, ED/UC visit, hospitalization and death. The estimated effectiveness against non-mortality outcomes is generally low, and lower than other estimates, including one from the same health system, though it studied a shorter time period. The manuscript is generally well written and the methodology generally appropriate to the chosen study design.

I do have one major and several minor some comments:

Major comment:

The outcomes being events within a specific time frame of a positive test, without any indication that the encounter was due to COVID-19 or even somewhat associated with respiratory illness is a concern. Our own work has shown that VE against hospitalizations verified to be due to COVID is generally somewhat higher than VE against hospitalizations with a code for acute respiratory infection, but not necessarily due to COVID. This study including encounters regardless of cause is likely to dilute the effect of vaccination by adding noise to the data. I suspect that this is as much of a reason for the lower VE in this study compared to TND studies using ARI-associated encounters as differences in follow-up time.

Response: We changed the primary outcomes to ED/UC visits or hospitalizations occurring from -1 to +10 days from the test-positive date and with an associated code for COVID-19 ((U07.1, U07.2, M35.81, and J12.82, **Supplemental Table 5**). COVID-19 VE was slightly higher compared to looking at all ED/UC visits or hospitalization in that time frame, as shown in the Table 1 below.

For completion we report as sensitivity analysis VE based on the following outcomes:

- All ED/UC visits and hospitalizations within -1 to +10 days from the positive SARS-CoV-2 test – **Supplemental Figure 3**
- ED/UC visits and hospitalizations within -1 to +10 days from the positive SARS-CoV-2 test that had associated codes for acute respiratory infection – **Supplemental Figure 4.**
- ED/UC visits and hospitalizations within ± 1 day of the positive SARS-CoV-2 test that had associated COVID-19 codes – **Supplemental Figure 5.**

- **Table 1. Comparison of COVID-19 VE for ED/UC visits and hospitalizations when defined as any ED/UC visit or hospitalization occurring from -1 to +10 days from a positive test versus ED/UC visits and hospitalizations during the same time interval that also had an association COVID-19 code**

	VE for ED/UC Visit			
	60 days	90 days	120 days	232 days
-1 to +10 days from positive test AND COVID-19 codes	34.40% (27.34-41.16)	29.19%, (23.65-35.00)	25.71%, (19.92-30.88)	21.05% (14.22-27.21)
-1 to +10 days from positive test: any ED/UC visits	33.20% (25.91-39.58)	27.20% (21.06-32.37)	24.00% (18.23-28.99)	18.41% (12.00-24.29)
	VE for Hospitalization			
	60 days	90 days	120 days	232 days
-1 to +10 days from positive test AND COVID-19 codes	37.39% (20.83-49.61)	28.98% (15.19-40.51)	22.52% (10.02-31.72)	19.53% (6.56-30.10)
-1 to +10 days from positive test: any hospitalizations	35.23% (22.16-47.58)	27.12% (13.36-38.08)	21.49% (9.31-32.30)	18.25% (5.24-28.21)

Minor comments:

Abstract: What time point was used for the Aalen-Johansen estimates to calculate the overall VE? It is presented as “Over a mean follow-up of 172 days (range 97-232), VE was low...” which makes it seem like it would be at 172 days, but it seems more likely to be at 232 days, the end of follow-up. Please clarify. The overall VE was calculated with follow-up starting from day 10 after vaccination (or index date) and extending through April 12, 2025, which corresponded to a maximum follow-up of 232 days and a mean follow-up of 172 days.

This point comes up again in the results, line 115: “through the end of follow-up (16.60%, 95% CI 11.92-21.44), corresponding to a mean follow-up of 172 days.” Is the VE estimate from the Aalen-Johansen estimates at end of follow-up (232 days) or mean follow-up (172 days)? If this is just a place to list the mean follow-up time, I’d move that up to the participant characteristics paragraph.

Response: The Aalen Johansen estimator extends to the end of follow-up at 232 days. This was clarified throughout the paper.

Lines 125-126: It should be the number needed to vaccinate to prevent one infection, not one visit.

Response: Thank you! We corrected this typo.

Lines 142-143: I would say that there’s no statistically significant effectiveness against death in the younger age group, though the number of events in this group is really too small to be able to say anything definitive – the CI includes the VE estimate for those age 75+. Also, the lower CI of -314.81 seems a bit extreme – can the authors please verify it? If it is correct, what use is a VE estimate with a CI nearly 400% wide?

Response: The point we are making is that the VE CI crosses zero for persons 18-74 years of age. The lower CI of -314.81 is correct and is an artifact of the bootstrap caused by the small sample size. The wide CI is due to the small number of deaths in patients 18-74 and we believe this is worth showing as some readers/investigators may be interested.

Lines 170-173: With the VHA-based TND study following patients through November (about 90 days since initial vaccination), I would expect their VE to be very similar to the 60-day estimates from this study –

median time since vaccination in a TND study is not the same as follow-up time in a TTE. Differences this large seem more likely to be due to some bias due to the study methodology (like the aforementioned lack of requirement that the encounter be associated with COVID or ARI) or perhaps differences in VE against different strains circulating at different times of year. A sensitivity analysis stratifying by enrollment period could potentially show if VE is waning by calendar time as well as time since vaccination.

Response: We calculated VE separately for “early trials” 1-3 (including 309,451 matched pairs enrolled from 8/23/24 to 10/27/24) vs “late trials” 4-8 (including 209,299 matched pairs enrolled from 10/28/24 to 01/17/25). We calculated 60-day and 90-day VE so that all participants in both the early and late trials would have sufficient follow-up. We did not find lower VE in the “late trials” versus the “early trials” (**Table 2**). This suggests that the decline in VE with longer duration of follow-up since vaccination is not related to waning VE by calendar time

We described this new analysis in the Methods and Results section and included a table of “early trials” versus “late trials” as **Supplemental Table 5**. We included a short paragraph in the Discussion to explain how these results suggest that the waning VE over time since vaccination that we observed is unlikely to be related to the evolution of new predominant variants against which the KP.2 vaccine offered significantly less protection.

Table 2. Comparison of COVID-19 VE for “early trials” with enrollment from 8/23/24 to 10/27/24 versus “late trials” with enrollment from 10/28/24 to 01/17/25

Outcome	Participants per arm, n	KP.2 Vaccination		No KP.2 Vaccination		Vaccine Effectiveness (95% CI)
		Events, n	Cumulative incidence at the end of follow-up per 1000 persons (95% CI)	Events, n	Cumulative incidence at the end of follow-up per 1000 persons (95% CI)	
SARS-CoV-2 Infection						
Follow-up to 60-days						
Early Trials	328,590	714	2.18 (2.02-2.34)	858	2.81 (2.57-3.06)	22.58(14.03-30.08)
Late Trials	208,876	493	2.37 (2.16-2.58)	970	3.82 (3.49-4.12)	37.97(29.94-45.01)
Follow-up to 90-days						
Early	328,590	1,284	3.93 (3.70-4.13)	1,447	4.87 (4.49-5.19)	19.35(13.12-24.85)
Late	208,876	729	3.51 (3.25-3.75)	1,351	5.32 (4.91-5.68)	34.11(27.45-39.71)
SARS-CoV-2 Associated ED/UC visit						
Follow-up to 60-days						
Early	328,590	361	1.10 (0.99-1.22)	462	1.52 (1.35-1.71)	27.76(15.90-37.31)
Late	208,876	278	1.33 (1.18-1.50)	549	2.16 (1.92-2.40)	38.20(27.89-47.32)
Follow-up to 90-days						

Early	328,590	664	2.03 (1.87-2.18)	795	2.68 (2.41-2.95)	24.39(15.87-32.37)
Late	208,876	413	1.99 (1.80-2.18)	770	3.03 (2.73-3.32)	34.54(25.56-42.84)
SARS-CoV-2 Associated hospitalization						
Follow-up to 60-days						
Early	328,590	93	0.28 (0.23-0.35)	128	0.42 (0.34-0.53)	33.00(10.19-50.26)
Late	208,876	67	0.32 (0.25-0.41)	137	0.54 (0.43-0.67)	40.34(18.65-55.84)
Follow-up to 90-days						
Early	328,590	166	0.51 (0.43-0.59)	208	0.70 (0.58-0.84)	27.82(10.04-41.27)
Late	208,876	104	0.50 (0.41-0.60)	183	0.72 (0.58-0.88)	30.73(10.98-47.56)
SARS-CoV-2 Associated death						
Follow-up to 60-days						
Early	328,590	**	0.01 (0.00-0.03)	19	0.07 (0.04-0.12)	86.08(55.36-100.00)
Late	208,876	**	0.02 (0.01-0.06)	13	0.05 (0.03-0.10)	54.32(-21.56-90.01)
Follow-up to 90-days						
Early	328,590	**	0.02 (0.01-0.04)	27	0.09 (0.05-0.15)	76.87(51.53-93.58)
Late	208,876	**	0.04 (0.02-0.08)	25	0.10 (0.06-0.17)	61.90(12.70-86.46)

Lines 205-208 seem to suggest that, because this vaccine was not terribly effective, next year's vaccine also won't be. That's a bit of a stretch, though I do agree that the sum total of current COVID—19 vaccination research does suggest that next year's vaccine will also wane in effectiveness over time, so an updated vaccine technology may be appropriate to address this general shortcoming of current vaccines.

Response: We agree. We edited the statement to clarify that “Although VE of the 2025-26 vaccine formulations cannot be predicted, our results ...”

We also updated this section with information on the recent FDA approval in August of 2025 of the JN.1/LP.8.1-targeting vaccine for adults ages 65 years and older as well as for individuals ages 5 through 64 with at least one high-risk condition.

In the limitations I would suggest adding potential generalizability concerns, as VHA members are significantly different from the general population.

Response: We added the following sentence in the limitations section.

“VHA enrollees included in our TTEs were predominantly male with advanced age and frequent comorbid conditions; our results should be extrapolated to other populations with caution”.

In Figure 3 the time when the number at risk and cumulative incidence are presented below the plots is not labeled. I assume that it is 0, 60 120, 180, and maybe 240 (or 232) days. I'd appreciate a label or footnote to confirm.

Response: We added a footnote to Figure 3 clarifying:

“The number of participants at risk (and number of cumulative events) shown below the x-axis correspond to 0, 60, 120 180 and 232 days”.

I am unsure of the journal's policy on significant digits for estimates, but two places after the decimal for VE estimates seems overly precise – I think one place after the decimal should be sufficient.

Response: We cannot find specific guidance in the Instructions for Authors. We left results with 2 decimal places throughout for consistency and to enable journal editors to change as needed.

Reviewer #2:

The authors report a study, designed using the TTE framework, on COVID-19 vaccine effectiveness. The study is well performed and described. We have a few points of feedback.

We commend the authors on reporting the Number Needed to Vaccinate, this is a very intuitive measure for interpretation.

Throughout the paper, the authors state that they study 'long-term follow-up'. While this is a subjective measure, we would not consider less than a year 'long-term'.

Response: We wanted to emphasize long-term follow-up because this distinguishes our study from already published studies with significantly shorter follow-up and allowed us to study waning over time.

It is a surprising result that the estimated VE against SARS-CoV-2 associated hospitalization is much lower than against SARS-CoV-2 associated death. Most studies find a lower VE against infection, but VE against hospitalization and death are usually both high. As all cause hospitalizations were included, could the authors report something about the registered causes of the included SARS-CoV-2 associated hospitalizations, or, preferably, run some sensitivity analyses?

Response: We changed the primary outcome to be limited to ED/UC visits and hospitalizations within -1 to +10 days of the positive SARS-CoV-2 test that also had documentation of COVID-19 codes (U07.1, U07.2, J12.82, M35.81, Supplemental Table 6). This resulted in slightly higher estimates of VE against SARS-CoV-2 associated ED/UC visits and hospitalizations.

We would have chosen the per-protocol analysis with censoring of the pair when the unvaccinated gets vaccinated as primary analysis, as this prevents part of the bias due to depletion of susceptibles, at least it keeps the vaccinated and unvaccinated in covariate balance. However, as the results hardly differ, we assume the adjustment in the primary analysis suffices.

Response: Thank you. We are presenting per-protocol analyses both by IPCW (main analysis in Figure 4) and by censoring the matched pair at the time of crossover (sensitivity analysis, Supplemental Figure 6). As the Reviewer points out the results are nearly identical.

How were positive tests within the first 10 days after the index date handled? Was the pair censored?

Response: As documented in the Statistical Analysis section “Primary analysis was limited to matched pairs in which both participants did not have a positive SARS-CoV-2 result during days 0-9 following the index date and remained alive, with follow-up beginning on day 10.” Out of the original 538,631 matched pairs, 537,466 remained alive and uninfected by day 10 and were included in the analyses.

Line 247: How likely is it that a VHA enrollee received a vaccination outside a VHA facility and how often are these recorded in the VHA EHR? As we read that older data were supplemented with CMS data. Can you say anything about the completeness based on past data?

Response: We captured COVID-19 vaccinations administered at VHA facilities or outside facilities and recorded in VHA EHR. Additionally, VHA participates in the CDC's Immunization Gateway, which provides almost real-time vaccination data from jurisdictional immunization information systems, and public and private vaccine providing organizations. Therefore, we did not require CMS data to capture vaccines administered during the enrollment period of the study. The comment about getting vaccine data from CMS before May 2024 refers to prior vaccines used to accurately ascertain prior vaccination status at the time of enrollment in the study. This was necessary because the CDC's immunization Gateway has been active for a couple of years and therefore it is not sufficient for capturing the prior vaccination histories

Minor points:

Line 64: from what age are children recommended vaccination?

Response: During the period of the study the Centers for Disease Control and Prevention (CDC) recommended that all persons ≥ 6 months of age receive a 2024-2025 COVID-19 vaccine dose. This is now clarified in the Introduction.

However, as of August 2025, FDA approved the JN.1/LP.8.1-targeting vaccines for all adults ages 65 years and older, and for individuals ages 5 through 64 with at least one high-risk condition. This is now clarified in the Discussion.

Line 74-75: "However, these studies estimated VE during a very short period after vaccination, which may overestimate VE over longer follow-up, if VE wanes over time." we would not call this an overestimation, it just estimates the short term VE which differs from the long term VE.

Response: We changed this sentence as follows: "However, these studies estimated VE during a very short period after vaccination, and therefore do not provide data on longer term VE, which may wane over time".

Line 81: a brief explanation of the TTE framework, with a reference, would be suitable here.

Response: The following statement was added: "TTE involves specifying the critical study design elements of an RCT (eligibility criteria, treatment strategies, treatment assignment, time zero, outcomes, causal contrasts and analysis plan) and explicitly attempting to emulate these elements in an observational dataset^{16,17}".

Line 85: "COVID-19 documented infection" I suggest to consistently use SARS-CoV-2 infection, as in the abstract

Response: We made changes throughout the paper to state "SARS-CoV-2" consistently throughout the paper when referring to the virus.

170 onward: is there any international literature on KP.2 VE?

Response: We repeated a comprehensive literature review and cannot find any other studies reporting the effectiveness of the KP.2 COVID-19 vaccine.

180: we would not state as strongly that there was less waning, as the uncertainty is high.

Response: We removed the reference to "less waning" over time for VE against death.

188-189: was KP.2 the dominant variant throughout the study period? How might this have influenced VE?

Response: There have been changes in predominant viruses during the period of the study, but all new variants that became dominant after KP.2 also descended from the same JN.1 lineage and had relative low number of spike protein amino acid differences. Our new additional analysis performed in response to the comments from Reviewer 1, did not show lower VE in persons vaccinated in late trials (October to January) than persons

vaccinated in early trials (August to October), This suggests that the decline in VE with longer duration of follow-up is not related to waning VE by calendar time related to new variants against which the KP.2 vaccine offered significantly less protection. We added these new results in the Methods, Results and Discussion sections.

219-220: besides missing at-home antigen test results, many infections will have remained untested, which can be added here as a limitation.

Response: We added that our study did not include infections that were not confirmed by any testing.

311: can you shortly elaborate on how the competing risk was incorporated in the analysis?

Response: We used a cause-specific Aalen-Johansen estimator which accounts for the presence of death as a competing risk in calculating the cumulative incidences. This was clarified in Methods.

Supplemental Table 1: How does reconfirming the matching criteria work? What happens if the match created at baseline is not an exact match at the index date anymore, or if the propensity scores are not the most similar anymore?

Response: Only eligibility was reconfirmed at the index date. We apologize that the heading of Supplemental Table 1 was confusing and we corrected that. This is also re-iterated in the Methods section:

Unvaccinated comparators had to remain eligible at the matched index date to be included, meaning they had to be alive, unvaccinated, and without a positive SARS-CoV-2 test.

“waning vaccinations after December”; the use of waning may be confusing here.

Response: We changed to “decline in vaccination rate after December”

Propensity scores: do you expect the number of prior COVID vaccinations to be a confounder?

Response: Potentially yes. Persons with higher number of prior vaccinations may be more likely to receive the KP.2 vaccine and may also be less likely to experience the outcome – if prior vaccinations continue to provide some protection.

Primary outcomes, right panel: For outcome 1, should this have the same text with ‘randomization date’ replaced by ‘index date’?

Response: Agree! We changed to “Same, except that outcomes are captured relative to the index date rather than the randomization date”.

Very minor:

103: participant > participants

260 withing > within

275 measure of 90-day mortality risk

315 form > from

318: as well as

Fig1: April 12, 2024 > April 12, 2025

Response: THANK YOU for picking up these embarrassing typos!

Reviewer #3 (Remarks to the Author):

Response: Thank you, and we are very impressed with and grateful for the review by the Early Career Researcher!

REVIEWERS' COMMENTS

Reviewer #1 (Remarks to the Author):

I would like to thank the authors for their work in updating the manuscript. They have appropriately addressed my concerns.

I am surprised that many of the point estimates for VE in the early-season vaccine recipients is lower than that for the late-season recipients. I wonder if this could be due to differences in the vaccinated population, with higher-risk and immunocompromised individuals more likely to be vaccinated shortly after the vaccine is available.

The Reviewer is correct that there were some systematic differences in early versus late-season vaccine recipients that may account for lower point estimates for VE in early versus late season trials. Specifically, early-season vaccine recipients were slightly older and had slightly higher comorbidity burden (as captured by the Care Assessment Needs [CAN] score). Another difference is that early-season vaccine recipients had much shorter interval since prior vaccination as shown in the Table below. Since, as we reported, VE against infections, ED/UC visit and hospitalization was greater in persons who had >365 days since last vaccination than in those who had 90-365 days since last vaccination, this may also partly explain the lower point estimates for the early-season vaccine recipients than the late-season recipients.

We included the table below as Supplemental Table 6, and added a sentence describing the differences between persons vaccinated in early trials versus late trials in the Results section, to provide context for readers interpreting differences in VE, as suggested by the Reviewer.

Table 1. Differences between participants in early (from 8/23/24 to 10/27/24) versus late (from 10/28/24 to 01/17/25) trials

	Early (N=65718 0)	Late (N=41775 2)	Overall (N=10749 32)
Received immunosuppressive or cancer medications in the last year, no. (%)	34931 (5.3%)	20990 (5%)	55921 (5.2%)
Age group (years), no. (%)			
18-64	145576 (22.2%)	117982 (28.2%)	263558 (24.5%)
65-74	201244 (30.6%)	128412 (30.7%)	329656 (30.7%)
≥75	310360 (47.2%)	171358 (41.0%)	481718 (44.8%)

	Early (N=65718 0)	Late (N=41775 2)	Overall (N=10749 32)
Care Assessment Need (CAN) score for 90-day mortality, no. (%)			
0-50	210302 (32.0%)	152536 (36.5%)	362838 (33.8%)
51-89	355336 (54.1%)	210528 (50.4%)	565864 (52.6%)
≥90	91542 (13.9%)	54688 (13.1%)	146230 (13.6%)
Time since most recent prior COVID-19 vaccine in days, no. (%)			
90-182	51786 (7.9%)	27078 (6.5%)	78864 (7.3%)
183-364	445700 (67.8%)	167106 (40.0%)	612806 (57.0%)
≥365	159694 (24.3%)	223568 (53.5%)	383262 (35.7%)

Reviewer #2 (Remarks to the Author):

We are happy with the revision and recommend it for publication.

Thank you!

Reviewer #3 (Remarks to the Author):

Thank you!